# Effect of Pig Synthetic Pheromones and Positive Handling of Pregnant Sows on the Productivity of Nursery Pigs

**DOI:** 10.3390/vetsci11010020

**Published:** 2024-01-03

**Authors:** Dimitri De Meyer, Ilias Chantziaras, Arthi Amalraj, Dominiek Maes

**Affiliations:** 1Unit of Porcine Health Management, Department of Internal Medicine, Reproduction and Population Medicine, Faculty of Veterinary Medicine, Salisburylaan 133, 9820 Merelbeke, Belgium; ilias.chantziaras@ugent.be (I.C.); arthi.amalraj@ugent.be (A.A.); dominiek.maes@ugent.be (D.M.); 2Vedanko, Knijffelingstraat 15, 8851 Koolskamp, Belgium; 3Unit of Veterinary Epidemiology, Department of Internal Medicine, Reproduction and Population Medicine, Faculty of Veterinary Medicine, Salisburylaan 133, 9820 Merelbeke, Belgium

**Keywords:** pigs, pheromones, nursery, performance

## Abstract

**Simple Summary:**

Weaning is one of the most important stress events in the life of pigs, increasing the risk for health problems and reduced performance. This study investigated the effect of pheromones on the performance of nursery pigs. In total, 24 batches of weaned piglets from a commercial farm (216 piglets per batch) were included. Half of the batches were treated, while the other half served as controls. Piglets of treated groups were exposed to a synthetic analog of the maternal pig appeasing pheromone (PAP) (SecurePig^®^, Signs, Avignon, France). The product consisted of a gel block containing the product and the pheromones diffused slowly in the room. The treatment with pheromones did not significantly influence the performance of the piglets (*p* > 0.05). The median values of average daily gain (318 vs. 305 g/day), feed conversion ratio (1.64 vs. 1.70), and number of antimicrobial treatment days (16.9 vs. 17.3 days) were numerically better in the nursery pigs exposed to the pheromones compared with the control groups. Mortality however was numerically higher in the treated groups (4.4 vs. 3.2%). In this farm, pigs exposed to the pheromone treatment during the nursery did not show a significant performance increase.

**Abstract:**

Weaning is one of the most important stress events in the life of pigs, increasing the risk for health problems and reduced performance. The release of pheromones in pig stables can be considered an environmental enrichment and alleviate the negative effects of weaning stress in nursery pigs. The present study investigated the effect of synthetic pheromones on the performance of nursery pigs. The effect of positive handling of sows in the farrowing house on the performance of the offspring in the nursery was also investigated. The study was performed in a commercial pig farm and included 24 batches of weaned piglets (216 piglets per batch). Half of the batches originated from sows exposed to positive handling. This implied that music was played from 6.00 a.m. to 6.00 p.m. from the moment the sows entered the farrowing house until weaning and that they were subjected to backscratching from the day they entered the farrowing unit the day of farrowing. During the nursery period, half of the batches were treated, and half served as controls. Piglets of treated groups were exposed to a synthetic analog of the maternal pig appeasing pheromone (PAP) (SecurePig^®^, Signs, Avignon, France). The product consisted of a gel block from which the pheromones were slowly released into the room. Different performance parameters were measured during the nursery period. Neither the sow treatment nor the treatment with pheromones significantly influenced the performance of the piglets during the nursery period (*p* > 0.05). The median values (95% confidence interval) of average daily gain, namely 318 (282–338) vs. 305 (272–322) g/day, feed conversion ratio, namely 1.64 (1.51–1.71) vs. 1.70 (1.57–1.75), and number of antimicrobial treatment days, namely 16.9 (9.6–25.0) vs. 17.3 (9.5–25.0) days, were numerically better in the nursery pigs exposed to the pheromones compared to the control groups. Mortality however was numerically higher in the treated groups, namely 4.4 (2.8–6.8) vs. 3.2 (0.9–4.2)%. Under the conditions of the present production system, pigs exposed to the pheromone treatment during the nursery did not show a significant performance increase.

## 1. Introduction

Olfaction plays a significant role in social recognition and discrimination of animals, and in establishing social hierarchy (Kristensen et al., 2001 [1]). Recognition of individuals will allow the animals to alter their behavioral response based on previous experience and help in sustaining a group structure. Piglets can detect maternal odors and can discriminate between mother and non-mother odors from 12 h after birth onwards (Faucitano and Schaefer, 2008 [2]). After birth, mammalian neonates including the pig must recognize the mother as this is essential for survival. This is particularly important for piglets as they need to compete with many littermates and find the nipple rapidly to survive (Guiraudie-Capraz et al., 2005 [3]).

During gestation, the mammalian fetus is exposed to various chemosensory stimuli (Smotherman and Robinson, 1987, Lecanuet and Schaal, 1996 [4,5]). This leads to the imprinting of the fetuses since newborns recognize the odors from amniotic fluid within 1 h after birth (Schaal and Orgeur, 1992 [6]), and are attracted by the odors from the maternal ventral skin (Morrow-Tesch and Mc Glone, 1990 [7]). This is also the case for pigs. Piglets use odor cues mediated by maternal fluids such as milk, colostrum, and amniotic fluid, for identifying the mother and teat position (Pageat and Teissier, 1998; Orgeur et al., 2002 [8,9]). Females and newborns develop a positive orientation toward these fluids (Lévy et al., 1983; Marlier et al., 1998, Schaal et al., 1994 [6,10,11]). This process involves nasal chemoreceptors, as flushing an anesthetic into the nose of newly born piglets strongly impaired their ability to locate a teat and initiate suckling (Morrow-Tesch and McGlone, 1990 [12]).

The ethological basis of this is well known but little is known about the nature of these cues nor the potential carrier molecules in maternal fluids such as amniotic fluids, colostrum, and milk (Pageat and Teissier, 1998 [9]). The odor components are likely maternal pheromones that are species-specific and composed of different fatty acids. The putative maternal pheromone of sows is composed of six fatty acids in the following relative proportions (Pageat, 2001 [13]): hexadecanoic acid (C16:0, 35%), cis-9-octadecenoic acid (C18:1, 26%), (cis,cis)-9,12-octadecadienoic acid (C18:2, 22%), dodecanoic acid (C12:0, 8%) tetradecanoic acid (C14:0, 7%), and decanoic acid (C10:0, 2%). The same fatty acids in similar patterns were found using gas chromatography and mass spectroscopy in amniotic fluid, milk, and colostrum (Guiraudi-Capraz et al., 2005 [3]).

The binding site of these putative pheromones are olfactory binding proteins (OBPs) in the nasal and vomeronasal mucosae of piglets (Guiraudie et al., 2003 [14]) Four main OBPs belonging to the lipocalin superfamily (Flower, 1996; Sivaprasadaro et al., 1993 [15,16]) have been identified: Alpha-1-acid glycoprotein (AGP), Odorant-binding protein (OBP), Salivary lipocalin (SAL), and Von Ebner’s gland protein (VEG). The role of these OBP’s in odor discrimination is still under discussion but it has been shown that OBP’s that exhibit ligand specificity (SAL and VEG) can bind other ligands in vitro (Guiraudie et al., 2003 [14]). It is suggested that binding site and shape are defined by the ligand and that OBP’s in several conformations are always present in the nasal and vomeronasal mucosae capable of binding multiple ligands with different affinities.

In older pigs, olfactory cues help in discriminating between familiar and unfamiliar conspecifics, and in the recognition of pen mates (Wells and Hepper, 2017 [17]).

Under natural conditions, piglets are weaned at 3 to 4 months of age, and it is a gradual process (Weary et al., 2008; Broom and Fraser, 2015 [18,19]). In intensive systems, however, piglets are often weaned between 3 and 4 weeks, or even at a younger age. It is an abrupt process with many stressors occurring at the same time, e.g., separation from the sow, changes in housing and feed, and mixing of piglets from different litters. Therefore, it is considered one of the most critical and stressful periods in the life of piglets (Martínez-Miró et al., 2016 [20]), and it may result in the development of psychobiological disturbances and welfare problems (Faucitano and Schaefer, 2008 [2]). Weaning causes an elevation in stress hormones (Held and Mendl. 2001 [21]), which may lead to a lower immune response and increased susceptibility to diseases (Kanitz et al., 2002 [22]). While establishing social hierarchy, fighting and aggressive interactions among pigs can occur in some farms and impair animal welfare and performance.

The European Union Directive 2008/120/EC [23] stresses the need for environmental enrichment to improve pig welfare (Vanheukelom et al. 2012 [24]). The release of pheromones in the pig stable could be considered an environmental enrichment method. Odor masking agents (OMA) may block olfactory signals from other pigs and disrupt social recognition causing temporary cessation of aggression (Kristensen et al., 2001; Marchant-Forde and Marchant-Forde, 2005). Guy et al. (2009 [1,25,26]) showed that the synthetic pheromone (Suilence^®^, Ceva Sante Animale, Libourne, France) may reduce aggression and fighting behavior between pigs, and influence feed intake during the post-weaning period. McGlone and Anderson [27] found an increase in daily weight gain after using the product. The application of a synthetic Pig Appeasing Pheromone (PAP) (SecurePig^®^, Signs, Avignon, France) in a slow-release format after weaning showed an increase in exploratory behavior with reduced agonistic behavior among weaned pigs (Temple et al., 2016 [28]). These effects were no longer statistically different at 24 h post-mixing. However, the above-mentioned studies did not show statistically significant effects on live weight, growth rate, or feed conversion efficiency. 

Overall, data on olfactory behavior, sensory enrichment, and their effects on piglet welfare and productivity in farm animals is limited. This is particularly the case for the use of odors in commercial swine farms. Hence the study investigated the effect of exposure to synthetic pheromones on different performance parameters of nursery pigs. The study is a continuation of a previous study (De Meyer et al., 2020 [29]) in which positive effects were shown on the preweaning survival of piglets when sows in the farrowing unit were exposed to music and when backscratching was done on the sows once per day. Therefore, the effect of this positive handling of sows on the performance of the offspring in the nursery was also investigated in this follow-up study.

## 2. Materials and Methods

### 2.1. Animals, Housing, and Management 

This study was performed in a commercial pig farm located in West Flanders, Belgium, that practiced a 2-week batch production system. The genetics of the sows originated from a commercial breeding company (Pig Improvement Company, PIC, Hendersonville, NC, USA). The sows were inseminated using semen from Piétrain boars. One batch of sows, comprising 56 sows, farrowed every two weeks. From weaning until four weeks of gestation the sows were housed in individual crates at the insemination unit, after which they were moved to a group housing system. Sows were fed a commercial gestation ration throughout the gestation period. A commercial feed containing barley, wheat, and wheat bran was given for the first 28 days after insemination at a ratio of 2.7 kg per day. A pelleted feed using trickle feeders (one for eight sows) was given while they were housed in groups. This was also a commercial diet containing wheat bran, beet pulp, and wheat and they received 2.3 kg per day. When the sows entered the farrowing rooms, they received a transition feed containing wheat, barley, and wheat bran, 2.5 kg per day. From 4 days after farrowing until weaning the sows received a lactation feed containing wheat, barley, and maize, 3 times per day on a classical schedule that is meant to reach a maximum (at libitum) before weaning. Between weaning and insemination, the sows receive a special diet for heat stimulation containing wheat, barley, mais, and extracted roasted soya at libitum. Sows (parity ≤ two) were vaccinated with Porcilis coliclos ^®^ (MSD) and porcilis ART ^®^ (MSD) at six and three weeks before farrowing. Sows (parity ≥ three) received a single vaccination, three weeks before farrowing. 

All sows irrespective of parity were vaccinated with Circoflex^®^, Mycoflex^®^, and PRRS-flex^®^ (Boehringer ‘triflex’) every four months. Post-farrowing vaccines include porcilis Ery-parvo^®^ (MSD) and respiporc flu^®^ (IDT). Prophylactic anthelminthic treatment with Panacur^®^ (MSD) was done before farrowing.

The piglets had received a dose of iron injection, 1 mL/piglet (Ferraject^®^, Dechra) during the first 24 h of birth. Tail docking and teeth grinding were also performed on day 1. Three days prior to weaning, piglets were vaccinated with Porcilis^®^ PCV M. Hyo (MSD Animal Health, Boxmeer, The Netherlands) and Porcilis^®^ PRRS (MSD Animal Health, Boxmeer, The Netherlands). 

At the end of the lactation (approximately 21 days), piglets were moved to off-site nursery units with two types of barn systems; 30 piglets per pen (‘A’ system) or 15 piglets per pen (‘B’ system). In the nurseries, the piglets were raised on partly slatted floors, and partitions between pens allowed physical contact with piglets from neighboring pens. In the large pens (30 piglets per pen), there were 6 feeding places and 2 drinking nipples; in the small pens (15 piglets per pen), there were 3 feeding places and 1 drinking nipple. The air spaces of the nursery units were separated from each other, and the units were ventilated with a mechanical (channel) ventilation system. A delta tube hot water system provided heating. The farm followed an all-in all-out system, and a hygiene lock was provided for visitors. The study took place in 2018. Zinc oxide (Gutal^®^—Huvepharma) at 3100 ppm was added to the feed for the first 14 days after weaning. Amoxycillin trihydrate (Octacilline^®^ 697 mg/kg—Eurovet) at 20 mg/kg was added to the drinking water for 8 to 10 days post-weaning to control *Streptococcus suis* infection. Piglets received ad libitum dry feed and drinking water. 

### 2.2. Experimental Design

The present study is a continuation of a previous study in which the effect of positive handling of the sows (treatment T) in the farrowing unit was investigated (De Meyer et al., 2020 [29]). Music was played from 6.00 a.m. to 6.00 p.m. from the moment they entered the farrowing unit until weaning. Simultaneously, farm personnel backscratched all sows of the T group once daily for 15 s with both hands in the middle of the back, from the day they entered the farrowing unit to the day of farrowing. For sows of the control (C) group, no music was played, nor backscratching was performed. 

The study included weaned piglets from 12 sow batches: the sows of six batches had been treated, and the sows of six batches had not been treated. From each sow batch, 432 piglets were selected at weaning and randomly allocated to two weaning batches, namely a treatment (*n* = 216) and a control group (*n* = 216). The combination of sow treatment and treatment of the nursery pigs resulted in four different groups: TT (treatment of sow and piglets), TC (treatment of sow, control piglets), CT (control sow, treatment of piglets), and CC (control sow and piglets). In total, 5174 weaned piglets were included namely 24 batches of 216 piglets each. A sample size of 6 batches for each group (i.e., a total sample size of 12, assuming equal group sizes) allowed for a statistical power of 80% and a level of significance of 5% (two-sided), to detect a true difference between the treated and the control group of 10 g and assuming a pooled standard deviation of 6 g.

For logistic reasons, weaning pigs from 4 out of the 6 sow batches that had been treated were housed in the nurseries of the A system (30 pigs per pen), whereas pigs from 2 sow batches were housed in the B system (15 pigs per pen). Similarly, weaning pigs from 4 out of the 6 sow batches that had not been treated, were housed in the nurseries with the A system; weaning pigs from 2 of the 6 sow batches that had not been treated were housed in the nursery room with the B system. 

During the nursery period, piglets of groups TT and CT were exposed to a synthetic analog of the maternal pig appeasing pheromone (PAP) (SecurePig^®^, Signs, Avignon, France). The active components of the PAP include methyl caprate, methyl laurate, methyl myristate, methyl palmitate, methyl linoleate, and methyl oleate. 

According to the manufacturer, PAP prevents the negative effects of stress and optimizes the performance potential of the animals. One PAP gel block of 250 g was used per 25 m^2^. The blocks were hung up at a height of approximately one meter above floor level, allowing the pheromones to slowly diffuse over the different pens of a compartment during a period of six weeks after the package was opened. Control piglets (TC and CC) received no treatment. As control and treated pigs were housed in separate compartments, control animals were not exposed to the pheromones released in the compartments of treated pigs. 

### 2.3. Parameters of Comparison

#### 2.3.1. Weight of the Piglets and Average Daily Gain (ADG)

The weight of the piglets at the weaning and end of the nursery was determined at the batch level. The end weight was determined at 42 days for the first 6 batches (3 controls and 3 treated), and at 39 days for the remaining 6 batches (3 controls and 3 treated). The dates when the piglets died were recorded. The weight of the dead pigs was estimated based on the age of the animal and a visual assessment. The average daily weight gain (ADG) was calculated by dividing the difference between the end weight (the sum of the weights that reached the end of the nursery period and the weight of the piglets that died) and the starting weight, divided by the number of pig days lived by the group (the pigs that reached the end of the nursery period and the piglets that died during the nursery). 

#### 2.3.2. Feed Intake and Feed Conversion Ratio (FCR)

The feed intake was measured at the batch level. Feed silos were emptied at the end of every nursery period (approximately 6 weeks) and feed remaining in the silos was weighed to calculate the group feed intake. For each batch, feed consumption was calculated by subtracting the quantity of feed remaining in the silo at the end of one nursery group from the total amount of feed that was delivered into the silo for that nursery group. The feed conversion ratio (FCR) was calculated by dividing the feed consumption of the batch by the weight gain of the batch.

#### 2.3.3. Mortality

The date and the estimated weight of the dead pigs were recorded. No necropsies or additional laboratory testing were performed on the dead pigs.

##### Antimicrobial Use

The number of days the pigs received antimicrobial medication during the nursery period, was recorded.

### 2.4. Statistical Analyses

The ADG, FCR, mortality, starting- and end weight of the piglets, and the number of antimicrobial treatment days were compared between the four groups CC, CT, TC, and TT. The normal distribution of the data was assessed with Q–Q plots, skewness and kurtosis ratio results, and the Shapiro-Wilk test. All parameters, except for FCR, end-weight, and mortality, were assessed as normally distributed. FCR was transformed using the reflect and square root method.

Per the model, the effects of sow and piglet treatment were taken into account. Interactions between sow and piglet treatment were examined and if found to be below *p* < 0.1 were kept in the model. For the parameters end weight and mortality that were also not normally distributed, the median values in the different groups were compared using the non-parametric Kruskal-Wallis test. The *p*-values were adjusted for all pairwise comparisons using the Bonferroni correction.

In addition, for end weight and mortality, separate analyses were performed to assess the effect of sow treatment in the farrowing unit regardless of piglet treatment (Table A1) and the effect of piglet treatment regardless of sow treatment (Table A2). The median values in the two groups were compared using the non-parametric Mann-Whitney test.

The results were expressed as the mean (standard deviation SD) and the median (confidence interval CI) values. The confidence interval of the median was calculated using a standard method as described by Zar [30]. The data were analyzed using SPSS software (SPSS version 27^®^; IBM Corp, Armonk, NY, USA). A *p*-value of 0.05 or lower was considered significant for all analyses.

## 3. Results

No interactions between sow and piglet treatment were found to be *p* < 0.1, and as such, they were not included in the final models. The effect of positive handling in sows in the farrowing unit and/or pheromone treatment in nursery pigs on average daily gain (ADG), feed conversion ratio (FCR), weight gained during nursery, mortality, and number of antimicrobial treatment days of nursery pigs are shown in Table A3 with their estimated means. Overall, only small and numeric differences were found in the mean values for these parameters between the different groups. The highest ADG and the lowest number of antimicrobial treatment days were found in group TT, and FCR was lowest in group CT. For end weight and mortality, we compared the medians of all groups, and no statistically significant (*p* < 0.05) differences were found. End weight was highest in group CT and mortality lowest in group TC.

When looking at the effect of positive handling of the sows regardless of piglet treatment, piglets originating from sows that had received positive handling in the farrowing unit had a numerically lower median ADG, but also a lower FCR, mortality, and number of antimicrobial treatment days during the nursery period (*p* > 0.05) (Table A1). The weight at the start of the nursery was slightly higher (*p* < 0.05) in the pigs originating from sows that had been treated in the farrowing unit compared with the weight of the piglets from control sows (5.89 vs. 5.57 kg).

Alternatively, when looking at the effect of pheromone treatment in the nursery pigs regardless of positive handling of the sows, piglets that received pheromone treatment in the nursery performed numerically better for ADG, FCR, and antimicrobial treatment but the mortality was also higher (4.37 vs. 3.22%) (*p* > 0.05) (Table A2).

## 4. Discussion

Separation of the offspring from the sows at weaning is often accompanied by social and physical changes in the piglets. The resulting stress might result in aggressive behavior and cause changes in feeding habits resulting in poor weight gain (Martinez-Miró et al., 2016 [20]). To create a familiar environment for the piglets, a synthetic pheromone product that mimics maternal presence in nurseries was used as an environmental enrichment method. The effects of this kind of enrichment on piglet performance were studied. 

The present study however could not demonstrate significant improvements in the performance of the nursery pigs exposed to the pheromone treatment. Average daily gain, efficiency of feed utilization, and treatment incidence with antimicrobials were slightly, numerically better in the treated batches, but no statistically significant differences were found. In a previous experimental study using a similar pheromone (Suilence^®^, Ceva Sante Animale, France), McGlone and Anderson [27] found a significant improvement in ADG and FCR. In the latter study, the pheromone was applied to the feeder, or directly to the snouts of the piglets. In the present study, the product was used according to the recommendations of the manufacturer namely, a gel block containing the product was hung up in the rooms and the product was released in the environment. Another study using PAP reported a temporary improvement in the explorative behavior of weaned piglets, but the performance of the pigs was not investigated (Temple et al., 2016 [28]).

The effects of the pheromone treatment during the nursery were not significantly influenced by the treatment of the sows (exposed to music and backscratching) of these piglets in the farrowing unit. Positive handling of the sows in the farrowing unit only resulted in a slightly significant increase in the weaning weight of the piglets. This means that the positive effects of sow treatment observed during the farrowing period (De Meyer et al., 2020 [29]) did not result in better performance of the piglets post-weaning. 

The present study focused on performance parameters and did not include other parameters such as fighting or other behavior characteristics of the piglets (Temple et al. 2016 [28]). According to the farmer and the herd veterinarian, there were no major problems with aggression post-weaning or poor performance of the piglets. It is possible that significant and more pronounced benefits would be obtained in farms that have problems with fighting piglets in the period post-weaning. This warrants further investigation. As only one production system was included in the present study, care should be taken when extrapolating the results to other pig farms. However, the farm characteristics as well as the performance results of the piglets are representative of other pig farms in Belgium (Malik et al., 2021; Declerck et al., 2016; Griffioen et al., 2023 [31,32,33]), and different successive batches including many piglets were included.

## 5. Conclusions 

The treatment might have improved the welfare of the animals, but under the present conditions, pigs exposed to the pheromone treatment during the nursery did not show a significant performance increase independently of the sow treatment in the farrowing rooms.

## Data Availability

Restrictions apply to the availability of these data. Data was obtained from the farmer and are available with the permission of the farmer.

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
