# Peer review of "Effect of Pig Synthetic Pheromones and Positive Handling of Pregnant Sows on the Productivity of Nursery Pigs"

_vetsci, 2024, doi:10.3390/vetsci11010020_

Round 1

Reviewer 1 Report

Comments and Suggestions for Authors

An interesting paper and research project. In the experimental design, it would help to know approximately how often and for how long sows had their back scratched. I don't recall any mention of the ventilation system used in the weaner accommodation. Could this be a factor in limiting the dose of pheromones to the individual piglets? Perhaps some facilities provide a higher dose based on their design?

The legend for each of the Tables is very long, and repeats a lot of what is in the methods. This might be a requirement of the journal, but if not then I suggest restricting the heading to just the first sentence.

I also suggest that ADG be given to just one decimal place

In Table A2, where there are significant differences it is normal for every value in that row to be given a letter and not just the two that are different. Again, this might be a requirement of the journal.

Table A3 doesn't need a note at the bottom about significant differences as there were none that I could see.

Author Response

Hello,

Please see the attached response.

Kind regards,

Dimitri De Meyer

Reviewer 2 Report

Comments and Suggestions for Authors

1. row 76- Guy et al (2009) must to be Guy et al. (2009);

2. references must to be with Abbreviated Journal Name from line 311 to 353.

The study involved enough weaned sows and piglets (5174 weaned piglets) and to make a conclusion.

Data processing was done in accordance with the Experimental Design method.

Perhaps more parameters could have been used, e.g. sensitivity stress analysis from blood parameters (one number pigs per group...) it would be a stronger manuscript.

The use of showed that the synthetic pheromone may reduce aggression and fighting behavior between pigs, and influence feed intake during the post-weaning period and for live weight, growth rate, or feed conversion efficiency but not statistically significant.

Pigs exposed to the pheromone treatment during the nursery did not show statistically significant effects on live weight, growth rate, or feed conversion efficiency but in the part of improving the rearing conditions for the welfare of pig farming, it gave an effect. Therefore, this study may have an effect in providing welfare, but it doesn't always have a significant effect on growth and feed conversion ratio... 

Author Response

Hello,

Please see the attached response.

kind regards,

Dimitri De Meyr

Reviewer 3 Report

Comments and Suggestions for Authors

This manuscript looks at the effects of using a synthetic pheromone on performance in the nursery phase of piglets. Overall it's a well-written manuscript. The authors found no significant differences in piglet performance even though manufacturer claims on the product say it optimizes performance.

Methods:

pg 3 Ln 119 - is the water treatment standard protocol or was there an outbreak

pg 3 ln 126 - what length of time daily were sows scratched

pg 4 ln 153 - was the six week period how long a block lasted or did you have to replace blocks and the 6 week period was your study period for when pigs went to the finishing phase? If the latter, how long did a block diffuse. Is there a published half life of the particles in the air? Were any measurements taken to be sure it was detectable at all levels of the pen and for how long ? 

Discussion

With PAP being more about maternal pheromones, it may be more beneficial investigating a shortened exposure time just to the initial weaning stress (e.g. 72h) to see if it has a positive impact. Especially if maternal pheromones should be calming during that period of separation. 

The effects of a pheromone may also wear off as the olfactory receptors become adaptive. Weights could've been taken more frequently for example, and broken growth performance up into growth periods to see if there was an initial affect and if it wore off. 

The manufacturer claim is for stress and performance potential, but perhaps performance would be more significant in finishers when lean muscle is being put on opposed to the grower nursery phase. Also, there may be more stressors in finishing than nursery to take affect on the overall performance and temperament of pigs. 

Author Response

(The authors gave the same response as above.)
